# Physico-Chemical Properties and Deposition Potential of PM_2.5_ during Severe Smog Event in Delhi, India

**DOI:** 10.3390/ijerph192215387

**Published:** 2022-11-21

**Authors:** Sadaf Fatima, Sumit Kumar Mishra, Ajit Ahlawat, Ashok Priyadarshan Dimri

**Affiliations:** 1CSIR-National Physical Laboratory, New Delhi 110012, India; 2Academy of Scientific and Innovative Research (AcSIR), Ghaziabad 201002, India; 3Atmospheric Chemistry Department, Leibniz Institute for Tropospheric Research (TROPOS), Permoserstraße, 04318 Leipzig, Germany; 4School of Environmental Sciences, Jawaharlal Nehru University, New Delhi 110067, India; 5Indian Institute of Geomagnetism, Navi Mumbai 410206, India

**Keywords:** PM_2.5_, chemical composition, morphology, deposition potential, smog, health effects

## Abstract

The present work studies a severe smog event that occurred in Delhi (India) in 2017, targeting the characterization of PM_2.5_ and its deposition potential in human respiratory tract of different population groups in which the PM_2.5_ levels raised from 124.0 µg/m^3^ (pre-smog period) to 717.2 µg/m^3^ (during smog period). Higher concentration of elements such as C, N, O, Na, Mg, Al, Si, S, Fe, Cl, Ca, Ti, Cr, Pb, Fe, K, Cu, Cl, P, and F were observed during the smog along with dominant organic functional groups (aldehyde, ketones, alkyl halides (R-F; R-Br; R-Cl), ether, etc.), which supported potential contribution from transboundary biomass-burning activities along with local pollution sources and favorable meteorological conditions. The morphology of individual particles were found mostly as non-spherical, including carbon fractals, aggregates, sharp-edged, rod-shaped, and flaky structures. A multiple path particle dosimetry (MPPD) model showed significant deposition potential of PM_2.5_ in terms of deposition fraction, mass rate, and mass flux during smog conditions in all age groups. The highest PM_2.5_ deposition fraction and mass rate were found for the head region followed by the alveolar region of the human respiratory tract. The highest mass flux was reported for 21-month-old (4.7 × 10^2^ µg/min/m^2^), followed by 3-month-old (49.2 µg/min/m^2^) children, whereas it was lowest for 21-year-old adults (6.8 µg/min/m^2^), indicating babies and children were more vulnerable to PM_2.5_ pollution than adults during smog. Deposition doses of toxic elements such as Cr, Fe, Zn, Pb, Cu, Mn, and Ni were also found to be higher (up to 1 × 10^−7^ µg/kg/day) for children than adults.

## 1. Introduction

The presence of primary and secondary anthropogenic PM together with favorable meteorological conditions can form a thick layer of haze/smog. Major European cities have suffered from severe smog conditions due to high PM concentrations [1]. During autumn and winter seasons, various Hungarian cities have witnessed smog conditions due to the combined effect of higher PM concentrations, favorable weather conditions, and geographical location [2]. In the Czech Republic, smog episodes were characterized by the highest PM_2.5_ concentrations and organic compounds, poor dispersion conditions, lower temperature, and temperature inversion conditions [3]. The northern parts of India have also experienced severe haze/smog conditions due to increased PM emissions from a variety of emission sources, including transportation, industrial, residential energy usage, and biomass burning [4]. The variations in meteorological conditions during post-monsoon and winter seasons in north India are favorable for PM build-up due to lower temperature, higher relative humidity, lower wind speeds, and significant changes in wind directions, leading to lower visibility conditions during smog episodes [5]. Due to frequent crop-residue-burning activities taking place in the northwest Indian states such as Punjab and Haryana, air quality degrades in Delhi and nearby areas, which also extends up to other Indo-Gangetic Plains (IGP) regions of India such as Uttar Pradesh, Bihar, and West Bengal [5,6]. Higher concentrations of PM_2.5_ cause serious respiratory and cardiopulmonary diseases such as asthma, bronchitis, etc., in the residents of Delhi [7]. During episodic cases, these particles consist of a different chemical composition which leads to haze/smog conditions increasing these illnesses and allergic diseases [8]. Each year smog conditions in Delhi lead to emergency shut-down of schools and large open gatherings, interruptions/cancellation of transportation services such as flight/train/road traffic, cancellation of games such as cricket matches, and heavy losses to the economic sector [9]. During 2017, a severe smog event was observed in Delhi during the month of November (post-monsoon) due to major crop-burning activities which took place in agricultural areas of nearby states such as Punjab, Haryana, and Uttar Pradesh [10]. During Smog-2017, air quality degraded to severe condition marked by poor air visibility, which badly affected daily life of Delhiites.

Since chemical constituents of PM play a major role in determining its source identification, it is necessary to study variations in chemical signature of PM during episodic cases such as smog formation. Anthropogenic PM contains different constituents, including sulfates, nitrates, mineral dust, metals, organics, black carbon, fly ash, etc. [11]. Inorganic constituents cover up to ~70% of PM mass, whereas organic compounds constitute almost 30% of the fine particulate mass [12]. Smog is mainly contributed by fine PM such as PM_2.5_, and ~70% mass of PM_2.5_ is composed of carbon (C), nitrogen (N), and sulphur (S) [13]. In Delhi, higher PM_2.5_ concentrations with increased N and S constituents were observed during winter/post-monsoon seasons as a result of fossil fuels combustion and increased biomass-burning activities. Moreover, higher N and S species formation gives rise to increased PM_2.5_ concentrations by the process of gas-to-particle conversions/oxidation/destruction of the primary aerosol particles during transboundary pollutant transfer in IGP regions such as Delhi [14]. During smog conditions, various PM bound metals were found in increased concentrations such as N, S, Cl, K, Si, Al, Zn, Pb, Fe, Mn, and Na [15], which may cause airway injury and inflammation through Fenton reaction [16]. In addition, the presence of transition metals such as Cr, Fe, Zn, Ni, Cu, Cd, etc., increase the production of reactive oxygen species (ROS) which cause oxidative stress as a result of which cells and tissues are damaged, leading to inflammation [17]. Moreover, these transition metals also cause genotoxic effects [18]. Fine particles (PM_2.5_) consisting of Cu, Fe, and elemental carbon (soot) are found strongly associated with death from heart attack and Chronic Obstructive Pulmonary Disease (COPD) [19], whereas the presence of heavy metals such as Pd, Cd, and Hg in aerosol particles may adversely affect the central nervous system [20]. Chemical compositions of inhaled PM may also cause pro-inflammatory response in nervous tissues that lead to neurodegenerative diseases [21].

In addition, morphological parameters of PM also play a major role in aggravating the adverse health effects due to their fine sizes and non-spherical shapes [22]. When sharp-edged fine particles enter into our lungs, they reach the alveoli and get retained in the lung parenchyma [23]. The chemical composition of these fine-sized deposited particles may adversely affect the physiological and biochemical processes within our body once they reach the cell system of our body. After inhalation, PM gets deposited in various parts of the human lungs depending on their size ranges. PM with diameter <2.5 µm may reach the respiratory bronchioles and finally the alveoli, which are common sites for gaseous exchange within our body [24]. These particles can penetrate the alveoli and affect gaseous exchange processes, and due to their fine sizes, they may even reach the bloodstream and adversely affect human health [25]. Particles with diameter <1 µm with transition metals behave similarly to gas molecules, and, therefore, by the process of diffusion, they penetrate deep down to the alveoli and through systematic circulation processes, reach cells/tissues and get translocated there [26]. In addition, along with the bloodstream, they may even become a part of the circulation system [23] and cause damage to other organs, such as the brain [27]. Deposition potential of various sizes of particles including PM_1_, PM_2.5_, and PM_10_ are found variable in different parts of human respiratory tract such as the head, trachea–bronchial, and pulmonary regions. Studies on deposition potential of PM for different regions of human lungs were carried out at Dehradun city, India [28], Chennai, India [29], and Xi’an City, China [30], whereas studies on deposition doses of different elements were studied at Kanpur [31], Dhanbad [32], and Delhi, India [33], showing the significance of PM deposition in human lungs during inhalation.

Worldwide limited studies are available on variations in PM concentrations, chemical compositions [1,2,3,15], and adverse health effects of PM [8,34] during smog episodes. In addition, very few studies are available on morphology of particles during smog conditions around the globe [35,36]. Since, limited studies are available for smog episodes in different parts of India [4,6,37], and extremely limited studies are available in Delhi for variations in PM concentrations [38] and inorganic components of PM (C, N, S components only) [5] during smog, this study is very useful for providing detailed variations in PM’s elemental and organic compositions along with morphological variations during smog episodes in Delhi. In addition, we found no such study on exposure assessment for deposition doses of PM (using the MPPD model) and associated elements during smog episodes available in Delhi and other parts of India. Therefore, the present study will provide new insights for a holistic approach towards study of physico-chemical parameters of PM and health effects in terms of its deposition potential during smog conditions.

The present study addresses the following objectives:Variations in PM_2.5_ concentration and meteorological parameters pre-, during, and post-Smog Event-2017.PM_2.5_ analysis for elemental composition, organic functional groups, and morphology during Smog Event-2017.Variations in PM_2.5_ deposition potential and elemental deposition doses for different age groups pre-, during, and post- Smog Event-2017.

## 2. Study Area and Methodology

Delhi is a part of the IGP belt with an area of approx. 1485 km^2^ (latitude: 28°24′17″ N to 28°53′00″ N; longitude: 76°50′24″ E to 77°20′37″ E), predominantly surrounded by two Indian states, namely Uttar Pradesh and Haryana, where the former lies in the East, and the latter covers other directions sharing their boundaries with Delhi, together known as National Capital Territory (NCT) regions. As per the 2011 census, Delhi’s population was estimated to be more than 11 million [39]. Delhi lies in a semi-arid zone of India where meteorological conditions, including temperature, relative humidity, and rainfall greatly vary. Delhi has a typical humid subtropical IGP climate with hot summers generally affected by a frequent number of dust storms and dry and mild winter seasons, having significant smog/fog conditions.

### 2.1. Sample Collection

The sampling site selected for the study is CSIR-National Physical Laboratory (CSIR-NPL), which is situated in the central part of Delhi (Figure 1). The site characteristics include thick vegetation cover from one side and busy road from another side where heavy traffic can be seen every day. PM_2.5_ samples were collected every 24 h for pre-, during, and post-smog events during the year 2017 starting from 1 November 2017 to 15 November 2017; details are provided in Table 1. PM_2.5_ aerosol particles samples were collected using a high-volume (~16.67 L/min flow rate) air sampler (Model-APM-550; Envirotech^®^, New Delhi, India) placed on a rooftop at a height of 10 m from the ground, away from any other interferences, which is in accordance with an earlier study [40]. The samples were collected on Polytetrafluoroethylene (PTFE) filter paper (47 mm diameter; Pall^®^, New York, NY, USA) which were desiccated for 24 h to remove moisture and weighed before and after each sampling to obtain gravimetric data.

### 2.2. Individual Particles Collection and Measurement

To collect individual particles for morphological analysis, tin substrates (purity: >99%; size: 1 × 1 mm^2^; thickness: ~0.1 mm) with marked exposure sides were placed on PTFE filters during sampling, as discussed in Section 2.1. The collected individual particles were further characterized for physico-chemical properties. The scanning electron microscopy with energy-dispersive X-ray spectroscopy (SEM-EDS) imaging technique (SEM Model: ZEISS; EDS Model: Oxford Link ISIS 300; facility present at CSIR-NPL, New Delhi, India) was used to obtain SEM micrographs for morphological and chemical signatures of individual particles collected on tin substrates. SEM technique provides high resolution images (~3 nm) of individual particles, whereas EDS gives elemental composition (beryllium (Be) to uranium (U)) present in the sample. Similar methodology for individual particles sample collection were followed in other studies in Delhi [40,42], but to the best of our knowledge, we found very limited studies on morphology of particles in Delhi and other parts of India, especially during episodic cases such as smog.

### 2.3. Quantitative and Qualitative Measurement of Bulk PM_2.5_ Samples

The PTFE filters of PM_2.5_ samples were analyzed for inorganic and organic signatures using wavelength-dispersive X-ray fluorescence (WD-XRF) and open path-Fourier transform infrared spectroscopy (OP-FTIR), respectively. Since, OP-FTIR is a non-destructive technique, filter samples were first analyzed for organic functional groups using this technique, followed by WD-XRF for obtaining inorganic chemical signatures present in the filter samples.

Organic functional groups present in PM_2.5_ aerosol were analyzed using OP-FTIR (Model: Bruker IFS 125M; facility present at CSIR-NPL, New Delhi, India). The mid infrared band (MIR) used as a light source and the transmitted radiation were detected using a wide-band mercury cadmium telluride (MCT) detector continuously cooled with liquid nitrogen. Nitrogen purging was performed in the sample chamber during each analysis to remove moisture and CO_2_ interference in the samples. Organic functional groups in the wavenumber range of 500–4700 cm^−1^ were detected in the sample.

PM_2.5_ filter papers were analyzed using WD-XRF (Model: Rigaku ZSX primus; facility present at CSIR-NPL, New Delhi, India). During sample analysis, filter samples were exposed to the X-ray tube of XRF, which excites atoms present in the sample and produces photons of characteristic wavelength, which were detected by detectors providing elemental composition of the samples. Data of blank filters were subtracted as a background during inorganic and organic analysis of filter samples. To the best of our knowledge, limited studies [5] are available related to inorganic and organic composition of PM during smog episodes in Delhi.

### 2.4. Secondary Data Collection

The meteorological parameters (temperature, relative humidity, wind speed, and air visibility) data were provided by India Meteorological Department (IMD), New Delhi. The backward wind trajectories were used for studying probable source contributions from transboundary pollution transfer using a NOAA HYSPLIT trajectory model [43]. The sources of fire count data that confirmed biomass-burning activities include NASA’s FIRMS data [44] published in the Indian newspaper, The Hindustan Times, published on 09 November 2017 [45].

### 2.5. Deposition Potential Calculation

The deposition potential of PM_2.5_ was calculated using the multiple path particle dosimetry model (MPPD)-Version 3.04 [46]. The MPPD model gives output data for particle deposition in different regions of the human respiratory tract (HRT), e.g., head, trachea–bronchial (TB) region, pulmonary (P) region, and total (Head + TB + P) deposition. This model considers three types of particle deposition processes, including diffusion, impaction, and sedimentation for the calculation of deposition fraction with three principal input sections described as follows:(i)Airway morphometry: Out of eight different airway morphometry models, we selected “Yeh-Schum age-specific model”, which considers the different structure of lungs in relation to respective age groups. As asymmetric branching structure of human lungs greatly causes bias in both airflow and particle deposition in HRT, we selected this model, as it provides multiple path (all airways) particle deposition. Age groups (children and adults) that were selected for the study included 3-month, 21-month, 28-month, 3-year, 8-year, 14-year, 18-year, 21-year, and 30-year. Values for other input parameters in the airway morphometry category were set as default specific to the respective age category (e.g., functional residual capacity (FRC) and upper respiratory tract (URT) volume).(ii)Inhalation properties: This included input parameters such as aspect ratio, particle diameter, density, etc. The value of aspect ratio for PM_2.5_ were set as 1.3 [42,47], particle diameter for PM_2.5_ was set as 2.5 µm, and particle density for PM_2.5_ was set as 1.5 g/cm^3^ for the calculations [48].(iii)Exposure conditions: Two types of exposure conditions can be chosen in the model as constant and variable. For our study we selected ‘constant exposure’ for estimating 24-h PM depositions of a given concentration (mg/m^3^) at a constant rate. Other input parameters included acceleration of gravity (981.0 cm/s^2^); body orientation (upright); breathing frequency (per min); tidal volume (in mL); inspiratory fraction (0.5); pause fraction (0); and breathing scenario (nasal) for each age group.

Deposition potential parameters studies included deposition fraction; deposited mass rate (µg/min); and deposited mass flux (µg/min/m^2^) of PM_2.5_ for all age groups pre-, during, and post-Smog Event-2017. The MPPD model was used in previous reported studies [28,29]. However, we could not find many studies incorporating the MPPD model for PM deposition potential, especially during smog episodes in Delhi and other parts of India, which provides novelty to the present study.

### 2.6. Exposure Assessment Calculation for PM_2.5_ Associated Elements

Health risk associated with the exposure of elements present in ambient PM_2.5_ were calculated for 18 elements using the USEPA numerical model [49]. Average daily dose (µg/kg/day) of different elements present in PM_2.5_ were calculated for children and adults during different days of sampling for Smog Event-2017 using Equation (1) as given below:(ADDinh, µg/kg/day) = C × InhR × EF × ED/BW × AT × PEF(1)

Here,

C = Metal concentrations in PM_2.5_ (µg/m^3^);InhR = Inhalation rate (m^3^/day) (7.63 for adults and 20 for children);EF = Exposure frequency (365 days/year);ED = Exposure duration (24 year for adults and 6 year for children);BW = Body weight (70 kg for adults and 15 kg for children);PEF = Particle emission factor (1.36 × 109 m^3^/kg);AT = Averaging time for non-carcinogens (365 days/year).

Exposure assessment studies using the above USEPA numerical model have been carried out in earlier studies in Delhi [33] and other parts of India [31,32]. However, we could not find many detailed studies on exposure assessment of different elements during smog events in Delhi.

## 3. Results & Discussion

PM_2.5_ samples were collected every 24 h for pre- (1 November 2017), during (8 November 2017), and post- (9 to 15 November 2017) Smog-2017, which were further analyzed for gravimetric, inorganic, organic, and morphological parameters as follows:

### 3.1. Variations in PM_2.5_ Concentrations and Meteorological Parameters during Smog Event-2017

The variations in PM_2.5_ concentration pre-, during, and post- Smog-2017 are shown in Figure 2. During pre-smog sampling (1 November 2017) the PM_2.5_ concentration was reported as 124.0 µg/m^3^, which drastically increased up to 717.2 µg/m^3^ on the first day of Smog Event-2017 (8 November 2017) which was ~6 times higher than that of pre-smog PM_2.5_ concentration (Figure 2). The previous study over Delhi reported the average PM_2.5_ concentration during Smog Event-2016 as 793 (±27.8) µg/m^3^ [5]. The increased PM_2.5_ concentration during Smog-2017 was reported ~29 times and ~12 times higher than the 24 h average PM_2.5_ permissible limit set by World Health Organization (WHO) and National Ambient Air Quality Standards (NAAQS), India, which are 25 µg/m^3^ and 60 µg/m^3^, respectively. Such a higher concentration of PM_2.5_ may cause adverse health effects even in the short-term and aggravate respiratory diseases such as asthma, severely affecting sensitive communities such as children and older persons. Although, PM_2.5_ concentrations decreased on consecutive days after the smog event (9/10 November 2017) but were still higher than the pre-smog PM_2.5_ concentrations due to post-smog effects. The post-smog effects included pollution build-up coupled with favorable meteorological conditions, such as lower temperature, lower wind speed, and higher relative humidity (Table 2). Similar findings were reported for Smog-2016 where post-smog PM_2.5_ concentrations increased with the dominance of secondary formed particulate ammonium sulphate via gas-to-particle conversion along with favorable meteorological conditions [5]. On the post-smog day (11 November 2017), higher PM_2.5_ concentration of 459.0 µg/m^3^ was observed which decreased on further consecutive days of post-smog event period (13/14/15 November 2017) (Figure 2). The increase in PM_2.5_ concentration on 11 November 2017 may be due to secondary particle formation leading to increase in fine PM concentration. It has been reported that secondary PM_2.5_ formation takes places due to photochemical reactions among multiple emitted pollutants, which causes severe haze conditions as reported in many cities around the world [50,51]. In addition, increases in N and S species on 11 November 2017 (Table 3) confirm the formation of secondary particles, such as ammonium sulphate, as reported during Smog-2016 [5].

The variations in meteorological parameters, such as temperature, relative humidity (RH), wind speed, and air visibility conditions in pre-, during, and post-Smog-2017 are shown in Table 2. On the pre-smog day, wind speed and temperature were found to be higher, 2.36 m/s and 23.3 °C, respectively. Higher wind speed leads to dispersion of pollutants, whereas higher temperature causes breakdown of pollutants in the presence of sunlight, leading to lower PM_2.5_ concentration [52,53], as also found in the present study. On the contrary, during Smog Event-2017, both wind speed and temperature were comparatively lower and RH was higher (Table 2), favoring the accumulation of particulate matter. In the presence of higher RH, PM acts as a nucleus for the condensation of water vapor present in the air, forming a dense mass known as fog. When this fog is mixed with other pollutants, such as S and N species from burning, it leads to the formation of smog [5]. During the severe haze period, heterogeneous reactions become the major formation pathway of secondary aerosol particles under high RH conditions [51]. The presence of lower PM_2.5_ concentration on pre-smog day also instigated good atmospheric visibility of 1942 m, whereas during smog conditions, visibility greatly reduced up to 385 m, which was lowest among all days of sampling. This signifies that meteorological parameters also played a significant role in PM pollution build up during Smog Event-2017.

Fire count data on 09 November 2017 [45] and backward air mass trajectory on smog day, i.e., 08 November 2017 (Figure 3) revealed that biomass-burning activities took place in the last week of October and the first week of November, 2017, during which air parcels moved from these areas to Delhi. This trans-boundary PM transport along with favorable meteorological conditions contributed to the severe smog event during the year 2017. In addition, the Punjab Pollution Control Board reported 39,686 burning events that took place in Punjab after 15 October 2017 [53]. The estimated total biomass burnt during 2017 until the Smog Event-2017 was reported as 23 million tons altogether from adjoining areas of Delhi, including Punjab, Haryana, and Uttar Pradesh [10]. Every year biomass-burning events take place during October-November months in these areas to make room for winter crops, causing severe smog conditions during post-monsoon/winter season.

### 3.2. Variations in PM_2.5_ Associated Elements, Organic Functional Groups, and Morphology during Smog Event-2017

#### 3.2.1. Elemental Composition

The variations in PM_2.5_ associated elements during Smog Event-2017 are shown in Table 3. On pre-smog day, the higher PM_2.5_ associated elemental concentration (>1 µg/m^3^) were reported for N (10.0 µg/m^3^), Si (2.1 µg/m^3^), S (16.3 µg/m^3^), Ca (1.1 µg/m^3^), Cr (2.9 µg/m^3^), and Zn (1.6 µg/m^3^), whose concentrations drastically increased during the smog event (Table 3). During the smog period, the PM_2.5_-associated elemental concentrations were reported much higher for elements such as N (58.0 µg/m^3^), Na (1.0 µg/m^3^), Mg (1.1 µg/m^3^), Al (4.8 µg/m^3^), Si (12.3 µg/m^3^), P (1.1 µg/m^3^), S (94.2 µg/m^3^), K (10.5 µg/m^3^), Ca (6.1 µg/m^3^), Cr (17.0 µg/m^3^), Fe (13.8 µg/m^3^), Zn (9.3 µg/m^3^), Pb (2.9 µg/m^3^), Cu (1.2 µg/m^3^), Br (0.9 µg/m^3^), and Ti (1.1 µg/m^3^) (Table 3). Higher concentrations of elements such as N, S, Cl, K, Cr, Fe, Zn, Pb, Cu, and Br during the smog event showed the signature of biomass-burning activities [2]. The presence of higher Al and Si during the smog event may be attributed to dust transport from the Punjab area along with air parcel movement to Delhi, as well as local road dust suspension contribution. During the smog episode, higher Cu and Zn concentrations found may be associated with brake/tire abrasion or emitted from lubricating oil [54]. In addition, higher concentrations of Zn and Pb in the droplet mode may be emitted from traffic and industrial sources [55]. Major elements found during the smog episode are S and N, which basically form smog. These elements greatly contribute to secondary gas-to-particle conversion processes, such as conversion of particulate NO_3_^−^ and SO_4_^2−^ from gaseous NO_x_ and SO_x_ in the presence of high humidity and low photochemical activity, increasing fine PM concentrations [5]. PM-bound S contributes to the formation of secondary inorganic aerosols containing NH_4_NO_3_ and (NH_4_)_2_SO_4_ under smog conditions [56]. In addition, sulfate formation via oxidation of SO_2_ is catalyzed by the presence PM metal ions such as Fe during the smog phase, increasing haziness and thus reducing visibility [5]. Higher concentrations of the elements in the present study may also be attributed to lower wind speed along with higher source contribution during the smog episode (Table 2). Typically, three constituents were found contributing to haze conditions in Shanghai, including secondary inorganic pollution, dust, and biomass-burning constituents [57].

#### 3.2.2. Organic Functional Groups

The qualitative variations in PM_2.5_-associated organic functional groups during Smog Event-2017 are shown in Table 4. FTIR analysis (qualitative) of PM_2.5_ provided variations in 17 organic functional groups present in pre-, during, and post-Smog Event-2017 which were identified by using the National Institute of Standards and Technology (NIST) library. On the basis of the source/mechanism of formation, organic functional groups have been categorized as [58]:Biogenic functional groups (ether, carbohydrates, hydroxyl groups, amino acids, and amines functional groups);Oxygenated functional groups (carboxylic acid, aldehydes, ketones, esters, lactone, and acid anhydride);Aliphatic hydrocarbon functional groups (aliphatic CH, alkenes, methyl, and methylene functional groups), and,Aromatic hydrocarbon functional groups, etc.

Table 4 shows lowest absorbance of organic functional groups on pre-smog day, whereas increased concentration during smog conditions. Dominant organic functional groups (absorbance > 0.10) on pre-smog day included alkyl halides (R-I) > alkyl halides (R-Br) > esters ≥ phenol ≥ amino acids/amines > carboxylic acid > alkane and alkyls > organonitrates > ether (Table 4). Dominant organic functional groups (absorbance > 0.10) during smog conditions included alkyl halides (R-Br) > alkyl halides (R-I) > alkane and alkyls ≥ carboxylic acids > esters ≥ phenol ≥ amino acids/amines ≥ organonitrates > aldehydes ≥ ketones ≥ carbonyl groups > amide > alcohol ≥ ethers ≥ alkyl halides (R-F) (Table 4). The presence of dominant species during smog showed their contribution from fossil fuel combustion including gasoline- and diesel-powered vehicles (alkyne, alcohols, nitro-compounds), burning activities (alkyl halides (R-F), alkyl halides (R-Br)), biogenic emission (ether), and oxidation processes (esters) [58]. In addition, aldehyde and ketones found to be originated from biomass burning, and ketones may also be formed from alkane oxidation processes [59]. Amine and amide groups found are mostly biogenic in emissions [60]. The percentage (%) contribution of different organic functional groups are presented in Figure 4. Higher % contribution of species, such as alkyl halides (R-I), alkyl halides (R-Cl), amines, amides, aldehydes, and ketones, confirm biomass-burning and biogenic emissions are dominant sources during Smog Event-2017. The presence of dominant organic functional groups during smog are found to be similar to those found in the ambient PM samples affected by wildfires and wood-burning activities [36]. Even after the smog event, significant absorbance of organic functional groups was found during the post-smog period (08/09/10 November 2017). This may due to the fact that biomass burning produces a significant number of organic compounds (VOCs and carbonaceous matter), which after oxidation with hydroxyl (OH) radical, become further increased in concentration during transboundary pollution transfer [61]. In addition, during haze conditions, organic compounds contribute to particle growth processes which are driven by secondary aerosol formation processes and air mass origin [62]. The organic compounds also contributed from moderate to severe haze conditions reducing visibility conditions in northeast China [15].

#### 3.2.3. Morphology

SEM-EDS micrographs shows the presence of individual particles in both spherical and non-spherical shapes during Smog Event-2017 (Figure 5). During the studied smog episode, non-spherical particles were abundantly dominant, followed by some spherical/nearly spherical particles. Round, flaky structures, solid and irregular-shaped particles, carbon fractals, sticky aggregates, sharp-edged particles, and rod-shaped and rectangular particles were abundantly found during the smog event. A variety of elements with variable mass % were found associated with these particles, such as C, N, O, Na, Mg, Al, Si, S, Fe, Cl, Ca, Ti, Cr, Pb, Fe, K, Cu, Cl, P, and F. The presence of elements such as C, N, S, Mg, Cl, K, F, Cr, Pb, Cu, etc., shows signatures from biomass-burning activities. The chained and branched structures (carbon fractals) rich in elements such as C, N, O, Na, and Fe were found, which may have originated from combustion processes. Aggregated and agglomerated particles rich in elements such as C, N, O, Na, S, Pb, Cl, Fe, K, and Cu were found in dominance showing biomass burning signatures. The irregular and sharp-edged particles with C, O, Na, P, S, and K were also found which may cause injury to inner tissues of lungs after inhalation. In addition, during smog, higher mass % of toxic elements in some individual particles such as Cu (up to 70%), Pb (up to 32%), and Cr (up to 25%) shows greater concern in terms of human health during smog conditions in Delhi. Metals such as Cr and Cu are common toxicants to lungs, and Pb causes neurological disorders while present in PM [63]. Similar studies for morphology and chemical composition of individual particles were conducted at Delhi, India [42], Haryana, India [64], Jaipur, India [65], Kanpur, India [66], Los Angeles, USA [67], and New Mexico [68], but limited studies on morphology of particles are available for episodic cases such as dust storms [40] and smog in Delhi.

### 3.3. Variations in PM_2.5_ Deposition Potential during Smog Event-2017

The variations in PM_2.5_ deposition potential for different age groups including children and adults, pre- during, and post-Smog Event-2017 using MPPD model are shown in Table 5 and Table 6. The MPPD model provides output data for deposition of particles in different regions of human respiratory tract (HRT), e.g., the head, TB, P, and total deposition. Deposition fraction is the ratio of the number of aerosol particles of a specific size (e.g., for PM_2.5_ particle diameter = 2.5 µm) deposited in a specific respiratory airway to the number of the same size entering the overall respiratory tract. Deposition rate determines the number of particles deposited in HRT per unit time and is represented in units as µg/min, whereas deposition flux determines the number of particles deposited in HRT per unit time per unit area of human lungs and is represented in units as µg/min/m^2^.

The variations in PM_2.5_ deposition fractions for different age groups are provided in Table 5. Deposition fractions in the pulmonary region were found highest, followed by the head region for the 3-month, 21-month, 8-year, and 14-year age groups, which signify more deposition of fine particles in the alveolar region for these age groups (Table 5). Deposition fractions in the head region were found highest, followed by pulmonary region for the 28-month, 3-year, 18-year, 21-year, and 30-year age groups, which signify more deposition of PM_2.5_ particles in the outer respiratory tract region than the alveolar region for these age groups (Table 5). Lowest deposition fractions were found in TB region for all age groups (Table 5). This signifies that babies (3-month and 21-month) and growing age children (8-year and 14-year) are more susceptible to fine particles deposition in inner parts of human lungs, e.g., alveolar region with potential health effects. The values of deposition fraction in our study are found to be in accordance with another study carried out in Chennai city, India, as shown in comparative Table 5 [29].

The MPPD model outputs suggested that on pre-smog day, lower deposited mass rate and mass flux for PM_2.5_ were observed, which drastically increased during smog conditions and later decreased on the last day of post-smog (15 November 2017) (Table 6 and Table 7). Increase/decrease in deposited mass rate and mass flux were observed due to respective increase/decrease in PM_2.5_ concentrations. Similar to the deposition fraction, deposited mass rate (µg/min) was found to be highest for the pulmonary region followed by the head region in specific age groups (3-month, 21-month, 8-year, and 14-year), whereas found highest for the head region followed by pulmonary region for remainder of the age groups studied (28-month, 3-year, 18-year, 21-year, and 30-year) (Table 6). Lowest deposited mass rate was found in TB region for all age groups during pre-, during, and post-Smog Event-2017 but with different values (Table 6). In addition, much higher values of deposited mass rate (µg/min) and mass flux (µg/min/m^2^) during smog conditions for all age groups showed the higher extent of PM_2.5_ deposition during pollution episodes in Delhi (Table 6 and Table 7). In the present study (Delhi, India), the highest total PM_2.5_ deposited mass rates (µg/min) in different age groups were reported during the smog event as 0.10 (3-month), 0.19 (21-month), 0.18 (28-month), 0.22 (3-year), 0.43 (8-year), 0.51 (14-year), 0.62 (18-year), 0.64 (21-year), and 0.67 (30-year) (Table 6). Among all age groups, the highest total deposited mass rate (µg/min) was observed for 30-year-old followed by 21-year-old adults throughout pre-, during, and post-Smog Event-2017 (Table 6). The highest value of total deposited mass rate for 30-year-old adults during pre-smog event was observed as 0.67 µg/min, which drastically increased up to ~6 times as 3.90 µg/min during smog event, which shows quite higher deposition of PM_2.5_ during episodic cases such as smog (Table 6). This higher rate of PM deposition may cause blocking of the upper respiratory tract or head region when individual particles with higher surface area are inhaled and get deposited because in this study, highest deposited mass rates are found for the head region for specific age groups (28-month, 3-year, 18-year, 21-year, and 30-year). On the other hand, smaller particles may reach the alveoli and bloodstreams and become deposited in internal organs of our body such as mitochondria, cytoplasm, and cytoplasm-bound vesicles [25] because in this study, highest deposited mass rate are found for pulmonary region for some age groups (3-month, 21-month, 8-year, and 14-year). When deposited inside cell organelles, PM causes oxidative stress, inflammation, mitochondria-induced apoptosis, and can interfere with intracellular proteins, organelles, and DNA present inside the cell [69]. The present values of deposited mass rate during smog were found to be quite a lot higher than the previously reported MPPD model outputs in India. For example, highest PM_2.5_ deposited mass rate (µg/hr) in Chennai, India, were found as 4.7 × 10^−6^ (3-month), 1.2 × 10^−5^ (28-month), 1.4 × 10^−5^ (3-year), 2.8 × 10^−5^ (8-year), 4.3 × 10^−5^ (9-year), 4.8 × 10^−5^ (14-year), 4.7 × 10^−5^ (18-year), and 6.0 × 10^−5^ (21-year) [29]. Highest PM_2.5_ deposited mass rate (µg/hr) in Dehradun, India, were found as 9.1 × 10^−3^ (3-month), 0.09 (21-month), 0.03 (3-year), 0.08 (9-year), 0.09 (18-year), and 0.12 (21-year) [28].

Mass flux maps are shown for different age groups pre-, during, and post-Smog Event-2017 in Table 7. Highest mass flux (µg/min/m^2^) values were reported for 21-month-old, followed by 3-month-old children, whereas lowest values were reported for 21-year-old adults, which highlights that PM_2.5_ exposure effects in terms of mass flux are higher for children than for adults (Table 7). This is due mainly to the fact that 21-month-old and 3-month-old babies have lowest lung surface area, whereas 21-year-old adults have largest lung surface area among all the age groups studied (Table 7). Highest mass flux during smog conditions were reported for 21-month-olds as 4.7 × 10^2^ µg/min/m^2^, followed by 3-month-olds as 49.2 µg/min/m^2^,whereas lowest as 6.8 µg/min/m^2^ for 21-year-old adults, which signify ~69 times and ~7 times more mass flux deposition in 21-month-old children, compared with 21-year-old adults (Table 6). This showed newborn baby and young child groups are more vulnerable than adults to PM_2.5_ mass flux, especially under episodic cases such as smog, which raises serious concern for pollution management strategies to avoid pollution episodes. Highest PM_2.5_ mass flux (µg/h/m^2^) values in Chennai city, India, were reported as 0.12 (3-month), 0.05 (28-month), 0.08 (3-year), 0.11 (8-year), 0.43 (9-year), 0.06 (14-year), 0.06 (18-year), and 0.05 (21-years) (Table 6) [29].

### 3.4. Variations in Average Daily Dose of PM_2.5_ Associated Elements during Smog Event-2017

Variations in average daily dose (ADD; in µg/kg/day) of 18 PM_2.5_-associated elements for children and adults on different days of sampling during Smog Event-2017 are shown in Table 8 and Table 9. Higher ADD were found for children than adults during smog conditions and lowest on pre-smog day, showing more elemental deposition in children lungs during smog events than adults (Table 8 and Table 9). During smog, highest ADD (µg/kg/day) was reported for different elements such as S (1.81 × 10^−7^), K (2.01 × 10^−8^), Ca (1.17 × 10^−8^), Cr (3.27× 10^−8^), Fe (2.65 × 10^−8^), Zn (1.79× 10^−8^), and Si (2.37 × 10^−8^) for adults (Table 8). For children, highest ADD (µg/kg/day) during smog was observed as S (5.54 × 10^−7^), Al (2.81 × 10^−8^), Cl (2.28 × 10^−8^), K (6.15 × 10^−8^), Ca (3.57 × 10^−8^), Cr (1.0 × 10^−7^), Fe (8.09 × 10^−8^), Zn (5.48 × 10^−8^), Pb (1.71 × 10^−8^), and Si (7.24 × 10^−8^) (Table 9). These highest ADD values during smog conditions show potential health effects of these elements after lung deposition in children and adults (Table 8 and Table 9). ADD for potential toxic elements were reported in the order of Cr > Fe > Zn > Pb > Cu > Mn > Ni with their highest ADD concentrations during Smog Event-2017 having potential toxicity after lung deposition (Table 8 and Table 9). In addition, values of ADD for toxic elements for children during post-smog condition (15 November 2017) were found comparable to those of ADD for adults during smog conditions. This signify that even after the smog episode was over in Delhi, children were still vulnerable to the post-effects of episodic cases such as smog in terms of PM_2.5_ lung deposition and associated health effects. The ADD for children is found to be higher than ADD for adults in other studies, which shows more exposure of children to the elements than adults [70]. According to a study conducted in Delhi, in the case of human exposure to metals, individuals were found exposed most to Fe, Zn, and Co and least exposed to Cd, Cr, and Pb during normal (non-episodic) days [33].

## 4. Conclusions

In the present study, lower concentrations of PM_2.5_, elemental composition, and organic functional groups were observed during pre-smog conditions which drastically increased during Smog Event-2017 in Delhi. Pre-smog PM_2.5_ concentration was reported as 124.0 µg/m^3^ which drastically increased up to 717.2 µg/m^3^ (~6 times higher) during the studied smog episode. The meteorological conditions, such as lower temperature, low wind speed, and higher RH played a crucial role in PM_2.5_ accumulation, thereby, reducing the air visibility to a greater extent in Delhi during the smog. During smog conditions, significantly higher concentration of elements such as N, S, Cl, K, Cr, Fe, Zn, Pb, Cu, and Br and the presence of dominant organic functional groups in PM_2.5_ confirmed biomass-burning signatures along with local emission sources in Delhi. Non-spherical-shaped individual particles dominated, followed by some spherical/nearly spherical-shaped particles with round, flaky structures, solid and irregular-shaped particles, carbon fractals, sticky aggregates, sharp-edged particles, and rod-shaped and rectangular particles during Smog Event-2017.

The MPPD model outputs suggested that during Smog Event-2017 much higher rate of PM_2.5_ deposition potential (mass rate and mass flux) were reported than that of pre- or post-smog days. The highest deposition fraction and mass rate (µg/min) were reported for the alveolar region of the lungs, followed by the head region in some age groups of babies and children, whereas, for the head region, followed by alveolar region in the remainder of the age groups, including adults. This shows that mostly the baby and child groups were affected by particle deposition in the alveolar region, causing adverse health effects, whereas in adults and other age groups, most of the particle deposition took place in the head region, affecting upper respiratory tract more than the alveolar region. Mass flux (µg/min/m^2^) in 3-month-old babies was found to be highest, followed by 21-month-old children, which shows babies and children were the most vulnerable to PM_2.5_ mass flux during the smog period due to lower lung surface area. In addition, ADD (µg/kg/h) for PM_2.5_-associated elements were found higher for children than adults. ADD for elements with potential toxicity, such as Cr, Fe, Zn, Pb, Cu, Mn, and Ni, were also found higher for children during smog conditions followed by the post-smog period, showing potential adverse effects of these toxic elements to children, even after the smog event was over. On the basis of this study, it can be suggested that there is an urgent need for atmospheric scientists, meteorologists, and policy makers to contemplate more stringent mitigation policies to combat pollution episodes such as smog to minimize the adverse health effects to the sensitive population, such as babies and children, in Delhi.

## Figures and Tables

**Figure 1 ijerph-19-15387-f001:**
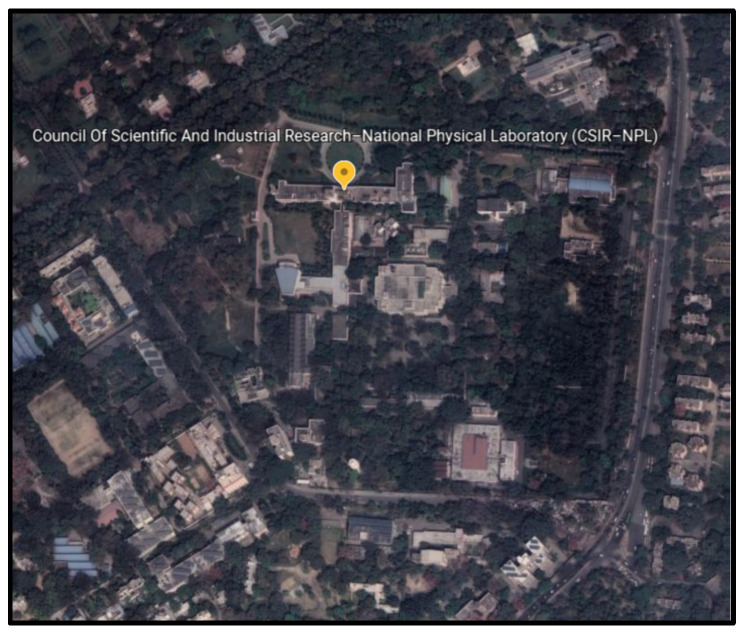
Sampling location CSIR-NPL at New Delhi, India [41].

**Figure 2 ijerph-19-15387-f002:**
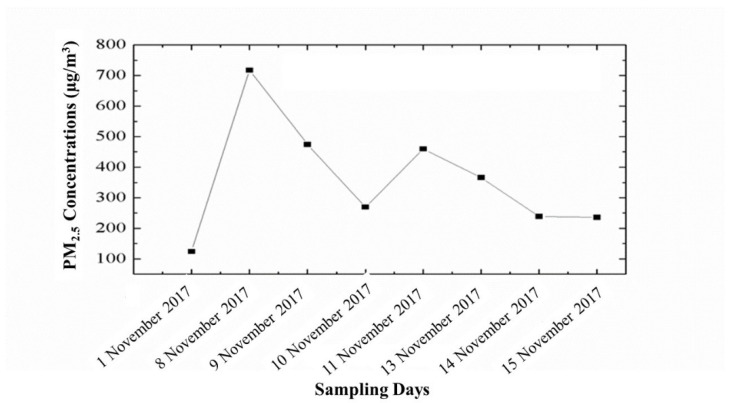
Variations in PM_2.5_ concentration (µg/m^3^) during Smog Event-2017.

**Figure 3 ijerph-19-15387-f003:**
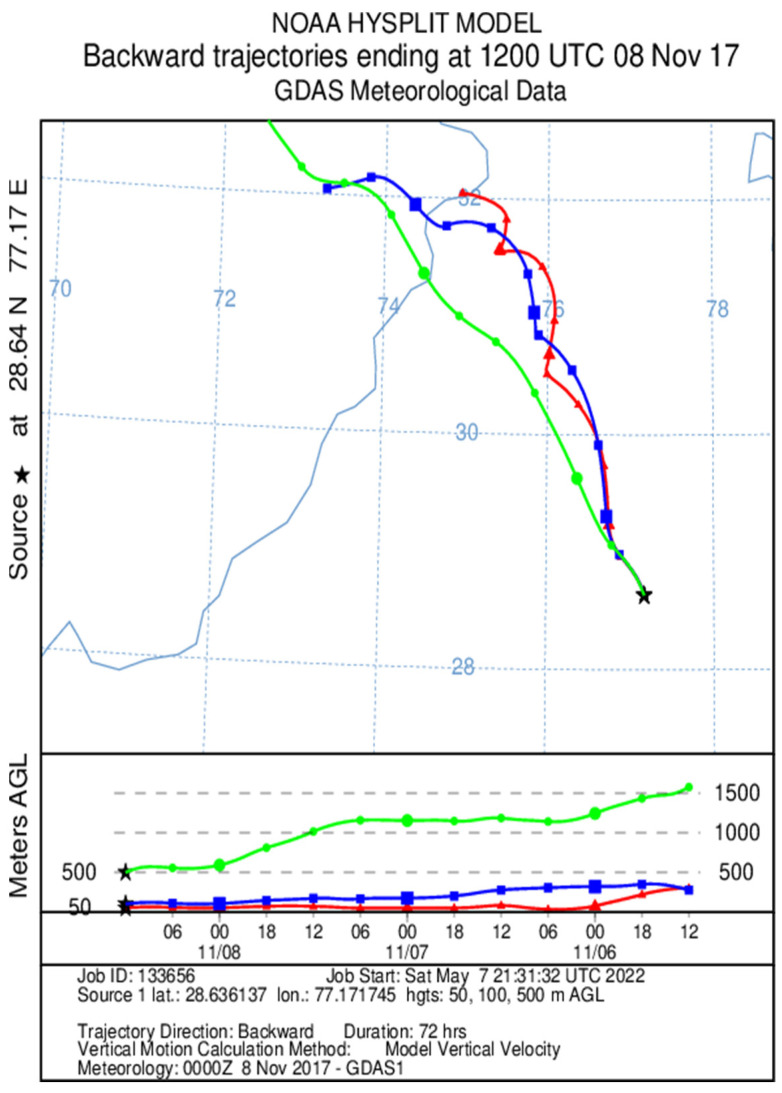
Backward air mass trajectory on 8 November 2017.

**Figure 4 ijerph-19-15387-f004:**
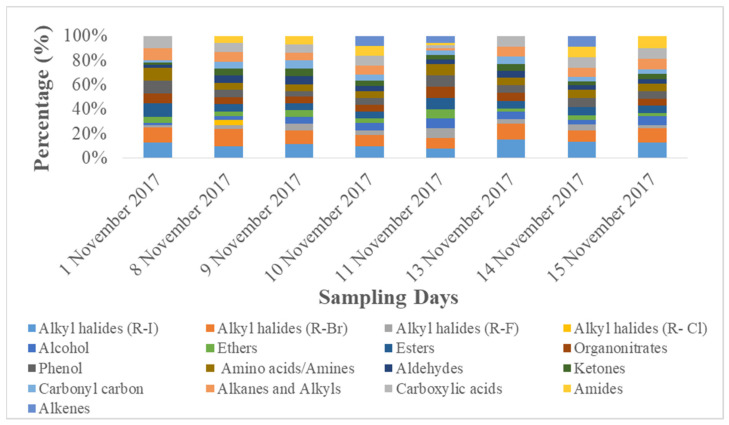
Percentage (%) contribution of different organic functional groups in PM_2.5_.

**Figure 5 ijerph-19-15387-f005:**
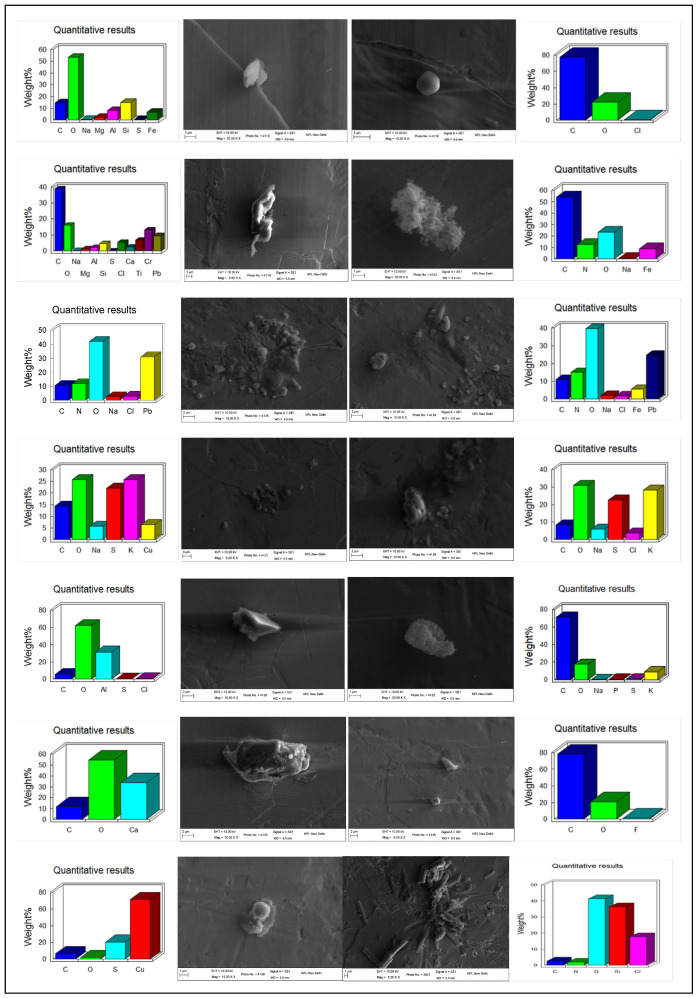
Morphology and chemical composition of individual particles during Smog Event-2017.

**Table 1 ijerph-19-15387-t001:** Details of sampling duration for PM_2.5_ samples collected during Smog-2017.

Sr. No.	Sampling Dates	Details
1	1 November 2017	Pre-smog
2	8 November 2017	Smog
3	9 November 2017	Post-smog
4	10 November 2017	Post-smog
5	11 November 2017	Post-smog
6	13 November 2017	Post-smog
7	14 November 2017	Post-smog
8	15 November 2017	Post-smog

**Table 2 ijerph-19-15387-t002:** Day-wise variations in meteorological parameters during Smog Event-2017.

Sampling Date	WS (m/s)	Visibility (m)	T (°C)	RH (%)
1 November 2017	2.36	1942	23.3	75.9
8 November 2017	0.71	385	19.8	81.9
9 November 2017	0.86	709	20.0	76.3
10 November 2017	0.60	940	19.8	76.1
11 November 2017	0.79	795	19.7	77.3
13 November 2017	2.07	942	19.1	79.9
14 November 2017	3.00	1123	20.0	75.5
15 November 2017	2.48	1806	19.4	75.0

**Table 3 ijerph-19-15387-t003:** Variations in PM_2.5_ associated elements (µg/m^3^) during Smog-2017.

Sr. No.	Elements	01 November 2017	08 November 2017	09 November 2017	10 November 2017	11 November 2017	13 November 2017	14 November 2017	15 November 2017
1	N	10.0	58.0	38.4	21.8	37.2	29.6	19.3	19.1
3	Na	0.2	1.0	0.7	0.4	0.7	0.5	0.3	0.3
4	Mg	0.2	1.1	0.7	0.4	0.7	0.6	0.4	0.4
5	Al	0.8	4.8	3.2	1.8	3.1	2.4	1.6	1.6
6	Si	2.1	12.3	8.1	4.6	7.9	6.3	4.1	4.0
7	P	0.2	1.1	0.7	0.4	0.7	0.6	0.4	0.4
8	S	16.3	94.2	62.3	35.4	60.4	48.1	31.4	31.0
9	Cl	0.7	3.9	2.6	1.5	2.5	2.0	1.3	1.3
10	K	-	10.5	6.9	3.9	6.7	5.3	3.5	1.8
11	Ca	1.1	6.1	4.0	2.3	3.9	3.1	2.0	2.0
12	Cr	2.9	17.0	11.3	6.4	10.9	8.7	5.7	5.6
13	Mn	0.2	0.9	0.6	0.3	0.6	0.5	0.3	0.3
14	Fe	2.4	13.8	9.1	5.2	8.8	7.0	4.6	4.5
15	Ni	0.1	0.7	0.5	0.3	0.4	0.4	0.2	0.2
16	Zn	1.6	9.3	6.2	3.5	6.0	4.8	3.1	3.1
17	Pb	0.5	2.9	1.9	1.1	1.9	1.5	1.0	1.0
18	Cu	0.2	1.2	0.8	0.4	0.7	0.6	0.4	0.4
19	Br	-	0.8	0.6	0.3	0.5	0.4	0.3	0.3
20	Ti	0.2	1.1	0.7	0.4	0.7	0.5	0.4	0.4

**Table 4 ijerph-19-15387-t004:** Variations in PM_2.5_-associated organic functional groups (in absorbance units) during Smog Event-2017.

Sr. No.	Functional Groups	01 November 2017	08 November 2017	09 November 2017	10 November 2017	11 November 2017	13 November 2017	14 November 2017	15 November 2017
1	Alkyl halides (R-I)	0.49	0.66	0.87	0.79	0.44	0.98	0.98	0.78
2	Alkyl halides (R-Br)	0.48	1.01	0.87	0.72	0.525	0.85	0.67	0.73
3	Alkyl halides (R-F)	0.06	0.24	0.41	0.28	0.46	0.22	0.39	0.14
4	Alkyl halides (R-Cl)	-	0.3	-	-	-	-	-	-
5	Alcohol	0.06	0.24	0.41	0.52	0.46	0.41	0.26	0.49
6	Ethers	0.21	0.24	0.41	0.28	0.46	0.16	0.26	0.14
7	Esters	0.41	0.42	0.44	0.45	0.54	0.41	0.52	0.37
8	Organonitrates	0.31	0.42	0.44	0.45	0.54	0.41	-	0.37
9	Phenol	0.41	0.42	0.302	0.45	0.54	0.41	0.52	0.37
10	Amino acids/Amines	0.41	0.42	0.44	0.45	0.54	0.41	0.52	0.37
11	Aldehydes	0.079	0.41	0.501	0.36	0.22	0.365	0.26	0.25
12	Ketones	0.079	0.41	0.501	0.36	0.22	0.365	0.26	0.25
13	Carbonyl carbon	0.079	0.41	0.501	0.36	0.22	0.365	0.26	0.25
14	Alkanes and Alkyls	0.36	0.54	0.47	0.62	0.1	0.55	0.52	0.53
15	Carboxylic acids	0.39	0.54	0.52	0.62	0.14	0.55	0.64	0.53
16	Amides	-	0.39	0.53	0.67	0.11	-	0.67	0.62
17	Alkenes	-	-	-	0.64	0.34	-	0.62	-

**Table 5 ijerph-19-15387-t005:** Comparative analysis of PM_2.5_ deposition fraction for different age groups.

Present Study		Age Groups								
Sr. No.	Deposition Fraction	3-Month	21-Month	28-Month	3-Year	8-Year	14-Year	18-Year	21-Year	30-Year
1	Head deposition fraction	0.24	0.25	0.29	0.28	0.27	0.26	0.42	0.41	0.47
2	TB deposition fraction	0.12	0.14	0.06	0.05	0.06	0.06	0.05	0.05	0.06
3	Pulmonary deposition fraction	0.30	0.29	0.21	0.28	0.40	0.33	0.28	0.31	0.20
4	Total deposition fraction	0.66	0.69	0.57	0.62	0.73	0.66	0.75	0.77	0.73
	Manojkumar et al., 2019 [29]									
5	Head deposition fraction	0.23	-	0.29	0.28	0.27	0.26	0.42	0.41	-
6	TB deposition fraction	0.12	-	0.06	0.05	0.06	0.06	0.05	0.05	-
7	Pulmonary deposition fraction	0.30	-	0.21	0.28	0.40	0.33	0.28	0.31	-
8	Total deposition fraction	0.65	-	0.56	0.61	0.73	0.65	0.74	0.77	-

**Table 6 ijerph-19-15387-t006:** PM_2.5_ deposited mass rate (µg/min) for different age groups during pre-, during, and post- Smog-2017.

Pre-smog		Age Groups								
Sr. No.	Deposition Potential	3-Month	21-Month	28-Month	3-Year	8-Year	14-Year	18-Year	21-Year	30-Year
1	Head deposited mass rate (µg/min)	0.04	0.07	0.09	0.10	0.16	0.20	0.35	0.34	0.44
2	TB deposited mass rate (µg/min)	0.02	0.04	0.02	0.02	0.03	0.05	0.04	0.04	0.06
3	Pulmonary deposited mass rate (µg/min)	0.04	0.08	0.07	0.10	0.24	0.26	0.23	0.26	0.18
4	Total deposited mass rate (µg/min)	0.10	0.19	0.18	0.22	0.43	0.51	0.62	0.64	0.67
Smog										
5	Head deposited mass rate (µg/min)	0.20	0.41	0.55	0.59	0.92	1.18	2.02	1.97	2.51
6	TB deposited mass rate (µg/min)	0.11	0.23	0.11	0.11	0.19	0.27	0.24	0.22	0.34
7	Pulmonary deposited mass rate (µg/min)	0.26	0.48	0.40	0.59	1.36	1.49	1.34	1.47	1.05
8	Total deposited mass rate (µg/min)	0.56	1.12	1.05	1.29	2.47	2.93	3.60	3.67	3.90
Post-smog										
9	Head deposited mass rate (µg/min)	0.07	0.14	0.18	0.19	0.30	0.39	0.66	0.65	0.82
10	TB deposited mass rate (µg/min)	0.03	0.08	0.04	0.04	0.06	0.09	0.08	0.07	0.11
11	Pulmonary deposited mass rate (µg/min)	0.08	0.16	0.13	0.19	0.45	0.49	0.44	0.48	0.34
12	Total deposited mass rate (µg/min)	0.18	0.37	0.35	0.42	0.81	0.96	1.18	1.20	1.28

**Table 7 ijerph-19-15387-t007:** Mass flux maps (µg/min/m^2^) for different age groups during pre-, during, and post-Smog-2017.

Age Groups	Normal Lung Geometry	Pre-smog	Smog	Post-smog
3-month	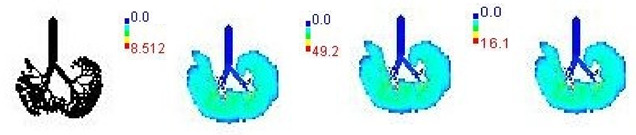			
21-month	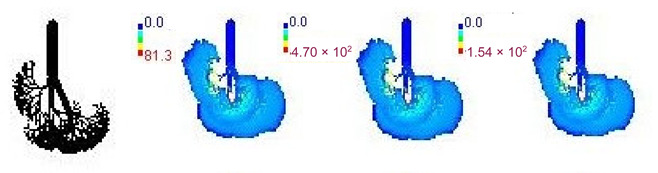			
28-month	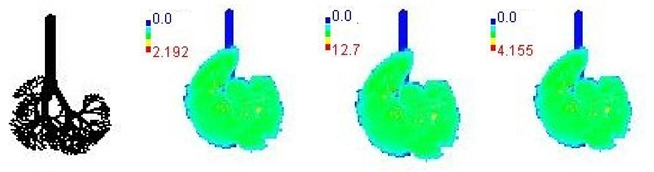			
3-year	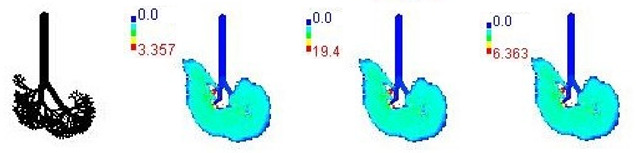			
8-year	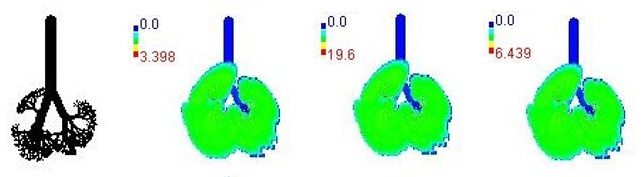			
14-year	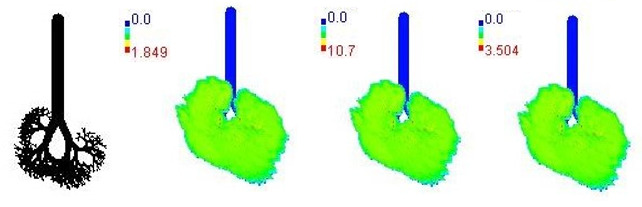			
18-year	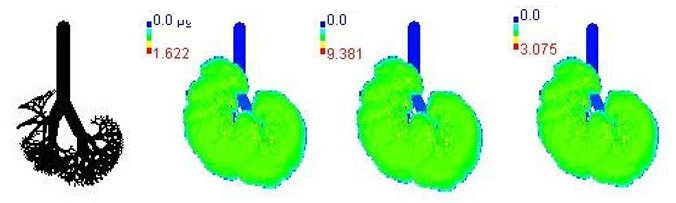			
21-year	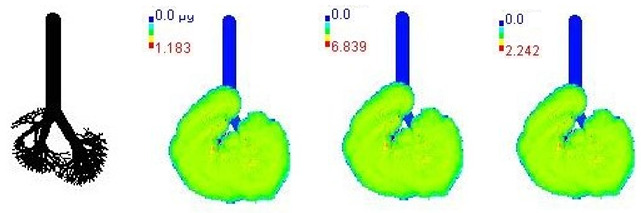			
30-year	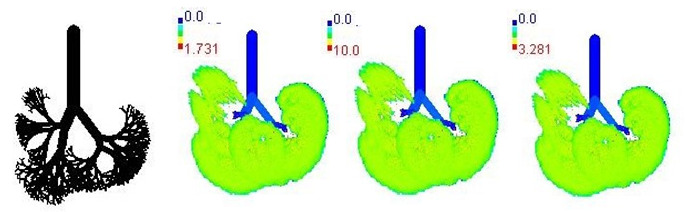			

**Table 8 ijerph-19-15387-t008:** Average daily dose (µg/kg/day) of different elements present in PM_2.5_ for adults during different days of sampling.

Adult	01 November 2017	08 November 2017	09 November 2017	10 November 2017	11 November 2017	13 November 2017	14 November 2017	15 November 2017
Na (µg/kg/day)	3.46 × 10^−10^	1.96 × 10^−9^	1.31 × 10^−9^	7.31 × 10^−10^	1.27 × 10^−9^	1.00 × 10^−9^	6.54 × 10^−10^	6.54 × 10^−10^
Mg (µg/kg/day)	3.46 × 10^−10^	2.06 × 10^−9^	1.37 × 10^−9^	7.69 × 10^−10^	1.31 × 10^−9^	1.06 × 10^−9^	6.93 × 10^−10^	6.73 × 10^−10^
Al (µg/kg/day)	1.6 × 10^−9^	9.19 × 10^−9^	6.08 × 10^−9^	3.46 × 10^−9^	5.89 × 10^−9^	4.69 × 10^−9^	3.06 × 10^−9^	3.02 × 10^−9^
P (µg/kg/day)	3.65 × 10^−10^	2.14 × 10^−9^	1.40 × 10^−9^	8.08 × 10^−10^	1.37 × 10^−9^	1.10 × 10^−9^	7.12 × 10^−10^	6.93 × 10^−10^
S (µg/kg/day)	3.13 × 10^−8^	1.81 × 10^−7^	1.20 × 10^−7^	6.81 × 10^−8^	1.16 × 10^−7^	9.25 × 10^−8^	6.03 × 10^−8^	5.96 × 10^−8^
Cl (µg/kg/day)	1.29 × 10^−9^	7.46 × 10^−9^	4.92 × 10^−9^	2.81 × 10^−9^	4.79 × 10^−9^	3.81 × 10^−9^	2.48 × 10^−9^	2.44 × 10^−9^
K (µg/kg/day)	-	2.01 × 10^−8^	1.33 × 10^−8^	7.56 × 10^−9^	1.29 × 10^−8^	1.03 × 10^−8^	6.69 × 10^−9^	3.48 × 10^−9^
Ca (µg/kg/day)	2.02 × 10^−9^	1.17 × 10^−8^	7.73 × 10^−9^	4.39 × 10^−9^	7.48 × 10^−9^	5.96 × 10^−9^	3.89 × 10^−9^	3.85 × 10^−9^
Cr (µg/kg/day)	5.66 × 10^−9^	3.27 × 10^−8^	2.16 × 10^−8^	1.23 × 10^−8^	2.10 × 10^−8^	1.67 × 10^−8^	1.09 × 10^−8^	1.08 × 10^−8^
Fe (µg/kg/day)	4.58 × 10^−9^	2.65 × 10^−8^	1.75 × 10^−8^	9.95 × 10^−9^	1.70 × 10^−8^	1.35 × 10^−8^	8.81 × 10^−9^	8.69 × 10^−9^
Ni (µg/kg/day)	2.31 × 10^−10^	1.33 × 10^−9^	8.85 × 10^−10^	5.00 × 10^−10^	8.46 × 10^−10^	6.73 × 10^−10^	4.42 × 10^−10^	4.42 × 10^−10^
Zn (µg/kg/day)	3.1 × 10^−9^	1.79 × 10^−8^	1.18 × 10^−8^	6.73 × 10^−9^	1.15 × 10^−8^	9.14 × 10^−9^	5.96 × 10^−9^	5.89 × 10^−9^
Mn (µg/kg/day)	3.08 × 10^−10^	1.75 × 10^−9^	1.15 × 10^−9^	6.54 × 10^−10^	1.12 × 10^−9^	8.85 × 10^−10^	5.77 × 10^−10^	5.77 × 10^−10^
Pb (µg/kg/day)	9.62 × 10^−10^	5.58 × 10^−9^	3.69 × 10^−9^	2.10 × 10^−9^	3.58 × 10^−9^	2.85 × 10^−9^	1.87 × 10^−9^	1.83 × 10^−9^
Cu (µg/kg/day)	3.85 × 10^−10^	2.21 × 10^−9^	1.46 × 10^−9^	8.27 × 10^−10^	1.42 × 10^−9^	1.14 × 10^−9^	7.31 × 10^−10^	7.31 × 10^−10^
Br (µg/kg/day)	-	1.62 × 10^−9^	1.08 × 10^−9^	6.16 × 10^−10^	1.04 × 10^−9^	8.27× 10^−10^	5.39 × 10^−10^	5.39 × 10^−10^
Si (µg/kg/day)	4.1 × 10^−9^	2.37 × 10^−8^	1.56 × 10^−8^	8.89 × 10^−9^	1.52 × 10^−8^	1.21 × 10^−8^	7.89 × 10^−9^	7.77 × 10^−9^
Ti (µg/kg/day)	3.46 × 10^−10^	2.02 × 10^−9^	1.33 × 10^−9^	7.50 × 10^−10^	1.29 × 10^−9^	1.04 × 10^−9^	6.73 × 10^−10^	6.73 × 10^−10^

**Table 9 ijerph-19-15387-t009:** Average daily dose (µg/kg/day) of different elements present in PM_2.5_ for children during different days of sampling.

Children	01 November 2017	08 November 2017	09 November 2017	10 November 2017	11 November 2017	13 November 2017	14 November 2017	15 November 2017
Na (µg/kg/day)	1.06 × 10^−9^	6.00 × 10^−9^	4.00 × 10^−9^	2.24 × 10^−9^	3.88 × 10^−9^	3.06 × 10^−9^	2.00 × 10^−9^	2.00 × 10^−9^
Mg (µg/kg/day)	1.06 × 10^−9^	6.29 × 10^−9^	4.18 × 10^−9^	2.35 × 10^−9^	4.00 × 10^−9^	3.24 × 10^−9^	2.12 × 10^−9^	2.06 × 10^−9^
Al (µg/kg/day)	4.88 × 10^−9^	2.81 × 10^−8^	1.86 × 10^−8^	1.06 × 10^−8^	1.80 × 10^−8^	1.44 × 10^−8^	9.35 × 10^−9^	9.24 × 10^−9^
P (µg/kg/day)	1.12 × 10^−9^	6.53 × 10^−9^	4.29 × 10^−9^	2.47 × 10^−9^	4.18 × 10^−9^	3.35 × 10^−9^	2.18 × 10^−9^	2.12 × 10^−9^
S (µg/kg/day)	9.58 × 10^−8^	5.54 × 10^−7^	3.66 × 10^−7^	2.08 × 10^−7^	3.55 × 10^−7^	2.83 × 10^−7^	1.85 × 10^−7^	1.82 × 10^−7^
Cl (µg/kg/day)	3.94 × 10^−9^	2.28 × 10^−8^	1.51 × 10^−8^	8.59 × 10^−9^	1.47 × 10^−8^	1.17 × 10^−8^	7.59 × 10^−9^	7.47 × 10^−9^
K (µg/kg/day)	-	6.15 × 10^−8^	4.07 × 10^−8^	2.31 × 10^−8^	3.94 × 10^−8^	3.14 × 10^−8^	2.05 × 10^−8^	1.07 × 10^−8^
Ca (µg/kg/day)	6.18 × 10^−9^	3.57 × 10^−8^	2.37 × 10^−8^	1.34 × 10^−8^	2.29 × 10^−8^	1.82 × 10^−8^	1.19 × 10^−8^	1.18 × 10^−8^
Cr (µg/kg/day)	1.73 × 10^−8^	1.00 × 10^−7^	6.62 × 10^−8^	3.76 × 10^−8^	6.42 × 10^−8^	5.11 × 10^−8^	3.34 × 10^−8^	3.29 × 10^−8^
Fe (µg/kg/day)	1.4 × 10^−8^	8.09 × 10^−8^	5.35 × 10^−8^	3.04 × 10^−8^	5.19 × 10^−8^	4.14 × 10^−8^	2.69 × 10^−8^	2.66 × 10^−8^
Ni (µg/kg/day)	7.06 × 10^−10^	4.06 × 10^−9^	2.71 × 10^−9^	1.53 × 10^−9^	2.59 × 10^−9^	2.06 × 10^−9^	1.35 × 10^−9^	1.35 × 10^−9^
Zn (µg/kg/day)	9.47 × 10^−9^	5.48 × 10^−8^	3.62 × 10^−8^	2.06 × 10^−8^	3.51 × 10^−8^	2.79 × 10^−8^	1.82 × 10^−8^	1.80 × 10^−8^
Mn (µg/kg/day)	9.41 × 10^−10^	5.35 × 10^−9^	3.53 × 10^−9^	2.00 × 10^−9^	3.41 × 10^−9^	2.71 × 10^−9^	1.77 × 10^−9^	1.77 × 10^−9^
Pb (µg/kg/day)	2.94 × 10^−9^	1.71 × 10^−8^	1.13 × 10^−8^	6.41 × 10^−9^	1.09 × 10^−8^	8.71 × 10^−9^	5.71 × 10^−9^	5.59 × 10^−9^
Cu (µg/kg/day)	1.18 × 10^−9^	6.76 × 10^−9^	4.47 × 10^−9^	2.53 × 10^−9^	4.35 × 10^−9^	3.47 × 10^−9^	2.24 × 10^−9^	2.24 × 10^−9^
Br (µg/kg/day)	-	4.94 × 10^−9^	3.29 × 10^−9^	1.88 × 10^−9^	3.18 × 10^−9^	2.53 × 10^−9^	1.65 × 10^−9^	1.65 × 10^−9^
Si (µg/kg/day)	1.25 × 10^−8^	7.24 × 10^−8^	4.78 × 10^−8^	2.72 × 10^−8^	4.64 × 10^−8^	3.69 × 10^−8^	2.41 × 10^−8^	2.38 × 10^−8^
Ti (µg/kg/day)	1.06 × 10^−9^	6.18 × 10^−9^	4.06 × 10^−9^	2.29 × 10^−9^	3.94 × 10^−9^	3.18 × 10^−9^	2.06 × 10^−9^	2.06 × 10^−9^

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
