# Peer review of "Physico-Chemical Properties and Deposition Potential of PM2.5 during Severe Smog Event in Delhi, India"

_ijerph, 2022, doi:10.3390/ijerph192215387_

Round 1

Author Response

The authors are grateful to the reviewer for their comments and suggestions. The justifications of the comments have been incorporated in the revised manuscript (Please see the attachment).

Reviewer 2 Report

Abstract should be cut in half.

Scientific or environmental background of Smog-2017 should be logically described first.

Devices and tools must be presented with brand, city, country etc.

Just under the line of 3. Results and discussion, the representatitveness of the sampling time and methodology should be clearly stated.

3.1. Variations in PM2.5concentrations -> make a space ...between PM2.5 and concentrations

Morphological observation with elemental analysis does not seem to be needed unless presenting different findings from 3.2.1.  It is tedious for readers to understand. 

Most of the discussions are quite general and easy to estimate.

More specific and particular findings including simple comparison with references are needed to make a better article.

Detailed description of Table 5 such as how to summarize and define would be better to be provided. (pls explain the MPPD a little bit)

Is Table 6 necessary?

Tables 7 and Figure 6 are duplicates.

line 505; why was higher deposition of PM2.5 found differing from other events? 

References should be reduced to less than a third.

 Although the significance of severe smog in a large city is presented, the reviewer feels a lack of scientific studies as an academic article.  

Author Response

(The authors gave the same response as above.)

Round 2

Reviewer 1 Report

I agree with the revisions for the manuscript and the rebuttals for those comments from reviewers done by the authors.  No more comments, the author has incorporated significant changes in the manuscript.

Author Response

Reviewer#1: General comment:

I agree with the revisions for the manuscript and the rebuttals for those comments from reviewers done by the authors.  No more comments, the author has incorporated significant changes in the manuscript.

Conclusion: I recommended this manuscript no further revision before publication in the journal.

Justification:

Authors are very grateful to the reviewer 1 for accepting the manuscript with no further changes.

Reviewer 2 Report

Introductory sentences should be omitted in Abstract.

Too general contents should be excluded in Introduction. 

Sampling site:
In order for the sample to be representative of the local air, it must be taken at a height with little influence from the ground. Thus, authors should mention the present sampling site was suitable.

Vivid purpose and novelty of this research are not clear. 

Authors should present and emphasize the difference of this work from other studies, particularly in Methodology or New findings.
